# Intensiometric biosensors visualize the activity of multiple small GTPases in vivo

Jihoon Kim[1], Sangkyu Lee[2], Kanghoon Jung[3], Won Chan Oh [3,4], Nury Kim[2], Seungkyu Son[1], YoungJu Jo[5], Hyung-Bae Kwon[3,6] & Won Do Heo[1,2,7]

Ras and Rho small GTPases are critical for numerous cellular processes including cell division, migration, and intercellular communication. Despite extensive efforts to visualize the spatiotemporal activity of these proteins, achieving the sensitivity and dynamic range necessary for in vivo application has been challenging. Here, we present highly sensitive intensiometric small GTPase biosensors visualizing the activity of multiple small GTPases in single cells in vivo. Red-shifted sensors combined with blue light-controllable optogenetic modules achieved simultaneous monitoring and manipulation of protein activities in a highly spatiotemporal manner. Our biosensors revealed spatial dynamics of Cdc42 and Ras activities upon structural plasticity of single dendritic spines, as well as a broad range of subcellular Ras activities in the brains of freely behaving mice. Thus, these intensiometric small GTPase sensors enable the spatiotemporal dissection of complex protein signaling networks in live animals.

[1] Department of Biological Sciences, Korea Advanced Institute of Science and Technology (KAIST), Daejeon 34141, Republic of Korea. [2] Center for Cognition and Sociality, Institute for Basic Science (IBS), Daejeon 34126, Republic of Korea. [3] Max Planck Florida Institute for Neuroscience (MPFI), Jupiter, FL 33458, USA. [4] Department of Pharmacology, University of Colorado School of Medicine, Anschutz Medical Campus, Aurora, CO 80045, USA. [5] Department of Physics, Korea Advanced Institute of Science and Technology (KAIST), Daejeon 34141, Republic of Korea. [6] Max Planck Institute of Neurobiology, Martinsried 82152, Germany. [7] KAIST Institute for the BioCentury, Korea Advanced Institute of Science and Technology (KAIST), Daejeon 34141, Republic of Korea. These authors contributed equally: Jihoon Kim, Sangkyu Lee, Kanghoon Jung. Correspondence and requests for materials should be addressed to H.-B.K. (email: hyungbae.kwon@mpfi.org) or to W.D.H. (email: wondo@kaist.ac.kr)

Working as a molecular switch, Ras and Rho small GTPases cycle between a GTP- and GDP-bound state to turn downstream signaling cascades on and off, respectively[1,2]. Accumulating evidence indicates that small GTPases have distinct activity profiles in space and time and are finely coordinated to determine proper cell fates in response to extracellular cues[3–5]. To understand the dynamic nature of small GTPase activity at a high spatiotemporal resolution, researchers have developed small GTPase biosensors, such as Förster resonance energy transfer (FRET) sensors[6]. FRET sensors have largely prospered in studies of small GTPases in diverse biological systems; however, several challenges remain for broad application of FRET sensors. First, the low sensitivity and efficiency of FRET sensors hamper in vivo application, especially for the visualization of protein activity at a micron scale. Although two-photon fluorescence lifetime imaging microscopy (2p-FLIM)[7] has been adopted to monitor active small GTPases at a subcellular level in complex multicellular environments such as organotypic brain slices, 2p-FLIM has not been directly applied to intact animal tissue. Second, as the FRET sensor innately requires donor and acceptor fluorescence, it practically limits the number of target signaling events that can be simultaneously analyzed in a single cell. Recently developed computational multiplexing approaches have compensated for this problem to some degree[8], but few references (e.g., cell membrane dynamics) for spatiotemporal alignments and correlative analyses of multiple signals collected from a number of samples restricts their widespread use. Third, considering the dramatic increase in the number of recently developed optogenetic techniques exploiting photoreceptors[9,10], it seems obvious that combinatorial uses of optogenetic manipulators and reporters will provide powerful means to dissect complex signaling networks with high spatiotemporal precision. However, the spectral overlap of cyan-yellow-paired FRET sensors with blue light-controllable optogenetic modules does not permit transient and local perturbation of target molecules under continuous monitoring of reporter activities[11].

To address these issues, here we present an intensiometric small GTPase biosensor that utilizes the dimerization-dependent fluorescent protein (ddFP). Green and red hues of our biosensors enabled multiplexed imaging of activity of multiple small GTPases in single cells. In addition, we demonstrated that red-shifted sensors are highly compatible with blue light-controllable optogenetic modules, thereby achieving spatiotemporal dissection of target activity under dynamic control of cellular functions. Applying to the mouse brain, we identified spatially distinctive activity profiles of Ras and Cdc42 in single dendritic spines and proved feasibility and effectiveness of our sensors for real-time monitoring of Ras activity in the brains of freely behaving mice at micron scale precision.

## Results

**Design of intensiometric Ras small GTPase biosensors.** The ddFP combines two quenched fluorescent protein-derived monomers: copy A and B. The copy A chromophore produces bright fluorescence when heterodimerized with copy B. Copy A can be spectrally diversified into green (GA) or red (RA) hue by mutations on multiple amino acids[12,13], copy B variants can bind to both GA and RA with similar or distinct affinities ($K_d = 3 \sim 40$ mM), implying possible production of multi-colored sensors[14]. Based on these properties, we designed intensiometric small GTPase sensors with a ddFP copy on both the small GTPase and effector domains. We hypothesized that activity-dependent binding of a small GTPase and effector would allow ddFP heterodimerization, thereby generating fluorescence (Fig. 1a). We initially examined basal fluorescence generated by inherent

dimerization of ddFPs. We measured fluorescence intensity in cells expressing GA with or without copy B. We used two variants of copy B, B1, and B3, which have different binding affinities to copy A ($K_d = 3$ μM, 40 μM for GA-B1, GA-B3, respectively)[14]. Compared with GA expression alone, co-expression of GA and each copy B increased basal fluorescence; the GA-B1 pair elicited a higher intensity than GA-B3. As both copies A and B were localized in the cytosol in this experiment, we examined whether spatial separation of GA and copy B could reduce fluorescence. Indeed, when GA was located in the plasma membrane and copy B in the cytosol, we observed significantly reduced fluorescence (Supplementary Fig. 1a, b). Thus, from the comparative analysis on the basal fluorescence, we decided to use GA-B3 pair rather than GA-B1 pair to minimize artificial signal and maximize dynamic range of sensor. Next, to examine whether we could utilize ddFP heterodimerization to represent small GTPase activity, we generated expression plasmids encoding the KRas mutant (the GTP- or GDP-bound form) and the Ras-binding domain of Raf (RBD$_{Raf1}$), labeled with GA, and B3 (hereafter called 'B') at their N-termini, respectively. RBD$_{Raf1}$ specifically binds to active Ras and their binding has been typically employed to generate various Ras biosensors[15,16]. We also tagged GA-KRas with iRFP682, a far-red fluorescence protein[17], to estimate the overall cell shape and relative expression levels of KRas in individual cells. When co-expressed with B-RBD$_{Raf1}$ in HeLa cells, iRFP-GA-KRas$^{GTP}$ showed much brighter green fluorescence than iRFP-GA-KRas$^{GDP}$ (Supplementary Fig. 2a), indicating that activity-dependent Ras-RBD interactions sufficiently induce ddFP heterodimerization and increased fluorescence. We evaluated the optimal configuration of fusion proteins by changing the relative positions or copy numbers of ddFP to Ras and RBD. Among the six combinations tested, N-terminal conjugations of GA and B to Ras and RBD, respectively, elicited the highest mean fold induction (~24.6-fold) of fluorescence (Supplementary Fig. 2b, c).

To monitor wild-type Ras activity, we designed a bicistronic vector encoding B-labeled RBD$_{Raf1}$ and GA-labeled KRas (B-RBD$_{Raf1}$-2A-GA-KRas) linked via a 2A peptide sequence[18], G-KRas (Fig. 1b). After expressing G-KRas in HeLa cells, we treated the cells with 50 ng ml$^{-1}$ epidermal growth factor (EGF) and observed a rapid increase (~2.8-fold) in G-KRas fluorescence intensity that saturated within three minutes, consistent with previous studies[15]. Treatment with a pharmacological inhibitor of the EGF receptor (gefitinib; 400 nM) reduced the augmented intensity, demonstrating that the Ras sensor fluorescence was specific to EGF signaling (Fig. 1b, c and Supplementary Movie 1). This reversible change implies that the sensor activity was mainly governed by the Ras-RBD interaction, which has three orders of magnitude higher affinity ($K_d = \sim 20$ nM)[19] than that between ddFP copies (Fig. 1d). Notably, G-KRas sensitively responded to even a pg ml$^{-1}$ concentration of EGF (Fig. 1e). To measure the detection limit of sensor, we performed signal-to-noise ratio (SNR) analysis for G-HRas in comparison with RaichuEV-HRas FRET sensor upon treatment of various EGF concentrations. As a result, we found that the detection limit for EGF by ddFP or FRET sensor was 0.011 ng ml$^{-1}$ or 24.946 ng ml$^{-1}$, respectively (Supplementary Fig. 3, and see Methods for SNR analysis in detail).

Given the structural similarity among Ras family members, we applied the same strategy to other small GTPases, such as HRas and NRas. G-HRas and G-NRas responded to EGF at a similar fold induction as that of G-KRas. In contrast, without either Ras or RBD$_{Raf1}$, the change in fluorescence was negligible (Fig. 1f). Given that copy B has a similar binding affinity to RA[14], we designed a red-colored KRas sensor (R-KRas) by replacing GA with RA in G-KRas. This sensor showed similar amplitude and fluorescence kinetics under EGF stimulation, suggesting the

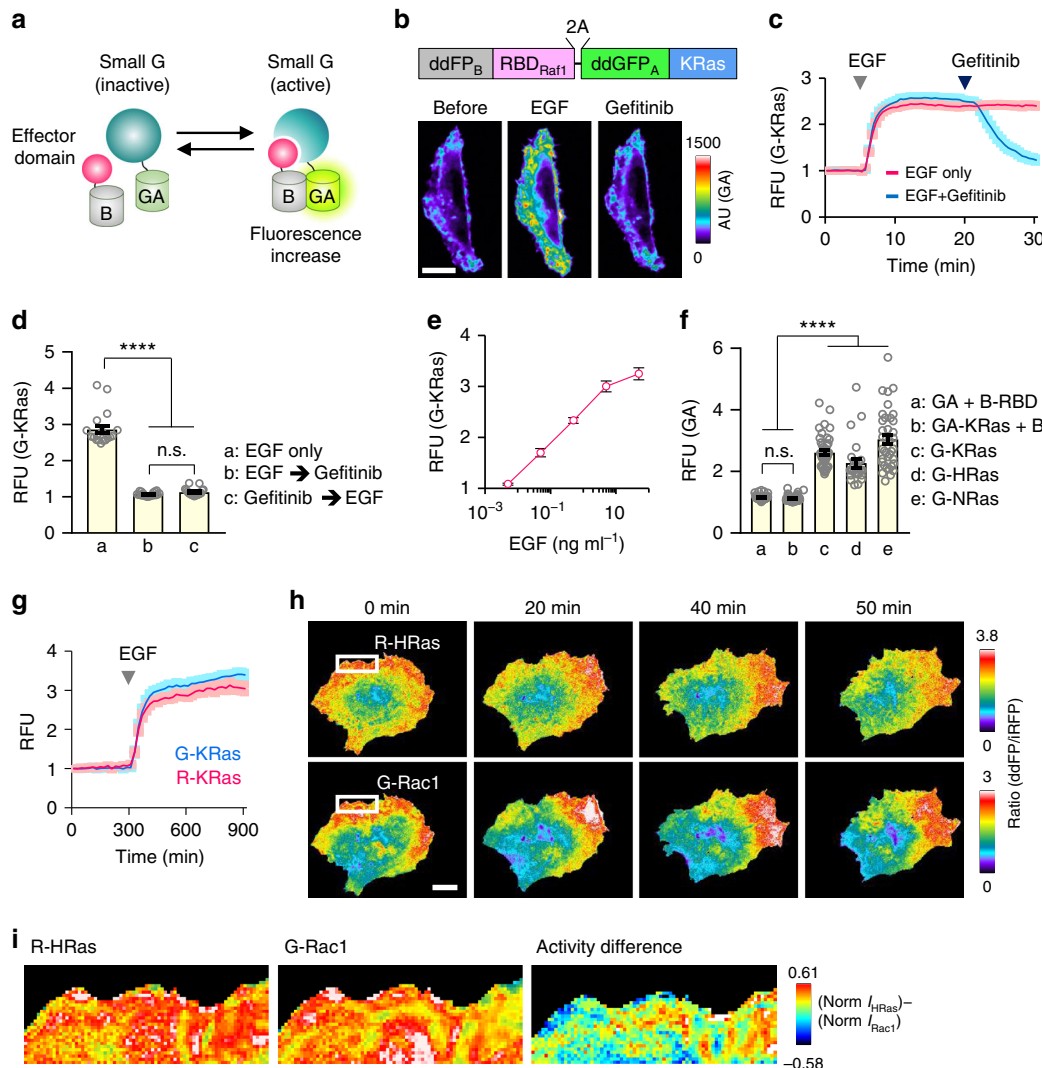

**Fig. 1** Development of intensiometric small GTPase biosensors. **a** Schematic of ddFP-based small GTPase sensor. **b** (top) Schematic depiction of KRas (G-KRas) sensor construct. (bottom) Fluorescence images showing Ras activity during sequential treatment of EGF (50 ng ml$^{-1}$) and EGFR inhibitor, gefitinib (400 nM). Color bar indicates range of G-KRas intensity. **c** Time-lapse graph represents reversible fluorescence change of G-KRas upon EGF or gefitinib treatment. $n = 77$ (blue), 40 (red). **d** Graph representing maximal fold-changes of GA-KRas intensity in HeLa cells. One way analysis of variance (ANOVA), $P < 0.0001$ for (EGF vs. gefitinib); $P < 0.0001$ for (EGF vs. pre-gefitinib + EGF); $P = 0.8948$ for (EGF+gefitinib vs. pre-gefitinib+EGF). $n = 19$, 20, 19; n.s., not significant. **e** Graph showing dose-dependent fold-changes of G-KRas sensor intensity upon EGF treatment. $n = 20$ for each concentration. **f** Quantification of maximal fold change of ddGFP-based Ras GTPase sensors upon EGF treatment. Each group of cells was co-transfected with plasmids as indicated. One way analysis of variance (ANOVA), $P < 0.0001$ for (**a** vs. **c**, **d**, **e**); $P < 0.0001$ for (**b** vs. **c**, **d**, **e**); $P > 0.9999$ for (**a** vs. **b**). $n = 32$, 42, 44, 27, 37; n.s., not significant. **g** Time-lapse graph showing dynamics of G-KRas and R-KRas sensor intensities upon EGF treatment. Images were captured at 20-s intervals. $n = 17$ (G-KRas), 18 (R-KRas). **h** Simultaneous imaging of HRas and Rac1 activities in MDA-MB-231 cell co-expressing R-HRas, G-Rac1, and Lyn-miRFP. Fluorescence ratio images showing Ras and Rac1 activity during cell migration. **i** (left, middle) Magnified images (area indicated by white boxes in **h**) showing subcellular distribution of R-HRas and G-Rac1 activities. (right) A color-coded image showing difference between normalized activities of HRas and Rac1 in subcellular regions. RFU: relative fluorescence unit; All scale bars, 20 μm; Error bars, s.e.m. Images or quantified data are representatives of multiple experiments ($N > 3$)

potential for simultaneous use of GA and RA (Fig. 1g and Supplementary Movie 2).

**Expansion of intensiometric biosensors to Rho small GTPases.** To expand the versatility of our platform, we next designed bicistronic expression vectors encoding red sensors of Rac1 and Cdc42 by employing CRIB$_{Pak1}$ as an effector domain, which is the most widely adopted module for generation of previous FRET sensors by its property of specific interaction with active Rac1 or Cdc42 (ref. [4,15]). To verify their functionality and specificity, we selectively activated small GTPases using rapamycin-induced

FKBP-FRB dimerization system for recruitment of catalytic domains of regulators to the plasma membrane[20] (Supplementary Fig. 4a). Upon translocation of the activators to the plasma membrane, R-HRas, R-Rac1, and R-Cdc42 showed a rapid and substantial increase in fluorescence; recruiting only FKBP did not affect fluorescence (Supplementary Fig. 4b). In addition, each sensor coupled with specific upstream activators (i.e., HRas-Sos1, Rac1-Tiam1, and Cdc42-Fgd1) (Supplementary Fig. 4c). We also confirmed that iSH2 (activator of endogenous PI3K) could increase fluorescence signals of all three sensors of R-HRas, R-Rac1, and R-Cdc42. Altogether, these results demonstrate

the sensitivity, specificity, and versatility of our sensors for visualizing Ras and Rho GTPase activities.

**Comparison of ddFP-based and FRET-based small GTPase biosensors**. In comparison with RaichuEV-HRas, the most widely used Ras FRET sensor[15], G-HRas exhibited a higher fold induction (~ 2.5-fold) of fluorescence upon EGF treatment. In addition, G-HRas showed significantly faster activation and deactivation dynamics than the RaichuEV sensor ($T_{1/2 \text{ on}}$ G-HRas: 80.59 ± 2.45 s, $T_{1/2 \text{ on}}$ RaichuEV-HRas: 260.64 ± 4.80 s, $T_{1/2 \text{ off}}$ G-HRas: 386.79 ± 8.46 s, $T_{1/2 \text{ off}}$ RaichuEV-HRas: 745.80 ± 22.23 s) (Supplementary Fig. 5a, b). We speculated that the different kinetics might be attributed to distinct intra- and intermolecular interactions of Ras and the effector domain in RaichuEV-Ras and G-Ras sensors, respectively. Thus, we tested an intermolecular Ras FRET sensor that consists of CyPet-HRas and YPet-RBD$_{Raf1}$. This sensor showed similar activation and deactivation kinetics as the ddFP-based sensor under EGF treatment (Supplementary Fig. 5c, d). This result could be explained by variations between the sensors in the accessibility of Ras to its regulators[21]. Next, as it is known that the local activity of small GTPase is important for various cell functions[4], we examined if the ddFP-based sensor correctly represents spatial distribution of protein activity by using the FRET sensor as a positive control. Analysis of signal distribution revealed that the localized activity pattern of R-HRas and G-Rac1 showed no significant difference in comparison with RaichuEV-HRas and RaichuEV-Rac1 respectively (Supplementary Fig. 6), indicating that replacement of reporter module from CFP-YFP pair to ddFPs does not cause any change or loss of spatial information of protein activity.

**Expression of ddFP-based sensor does not perturb Erk signaling**. We next examined whether expression of a ddFP-based Ras affected downstream ERK signaling. To analyze endogenous ERK activity in a single cell expressing the Ras sensor, we employed an ERK kinase translocation reporter (KTR)[21]. Quantification of the nucleocytoplasmic ratio of the ERK KTR sensor indicated that cells with or without Ras sensor showed little difference in basal ERK or EGF-stimulated activity (Supplementary Fig. 7). Therefore, the expression of the ddFP-based sensor does not perturb endogenous signaling under our experimental conditions.

**Visualization of multiple small GTPase activities in a single cell**. With the goal of directly measuring the activity of multiple small GTPases in a single cell, we developed ddFP-based sensors with spectrally separated hues. We co-expressed R-HRas, G-Rac1, and Lyn-miRFP in MDA-MB-231 cells. We used Lyn-miRFP as a membrane marker for normalizing the local fluorescence change caused by membrane fluctuations. Under random cell migration, we found that HRas and Rac1 were highly activated in the leading edge. Their activity profiles were positively correlated but showed slightly different spatial patterns under microscale comparison (Fig. 1h, i and Supplementary Movie 3). A previous study indicated that copy B can interchangeably bind to GA and RA under a certain proximity level[14], raising the possibility that copy B-fused effector might interact with a copy A-labeled non-target small GTPase. To examine this, we co-expressed MDA-MB-231 cells with R-HRas and GA-Rac1-IRES-Lyn-iRFP682 and monitored RA and GA signals over time. We observed a remarkable increase of RA signal from R-HRas sensor near the protruded area but no noticeable elevation of GA signal, indicating that binding of B-RBD$_{Raf1}$ to RA-HRas did not affect GA-Rac1 signal (Supplementary Fig. 8a). Similarly, when cells were expressed with G-Rac1 with

RA-HRas-IRES-Lyn-iRFP682, there was clear enrichment of GA signal at the area of leading edge but no detectable RA signal, further supporting the specificity of our sensors (Supplementary Fig. 8b). These results demonstrate the feasibility of using multiple ddFP sensors in the same intracellular environment without cross-reaction.

**High compatibility of red-shifted sensors with optogenetic modules**. A recently developed series of optogenetic tools utilizing photoreceptors provide fine-control of intracellular signaling in space and time[9,10]. However, owing to spectral overlap, blue light-excitable photoreceptors are not compatible with a cyan-yellow pair of FRET sensors[11]. We hypothesized that our red-shifted sensors could overcome these challenges. To examine this, we co-expressed R-HRas sensor with OptoFGFR1, an optogenetic module for activation of FGFR1 signaling[22], in MDA-MB-231 cells (Supplementary Fig. 9a). Upon brief illumination of blue light, we observed rapid and reversible activation of Ras ($T_{1/2 \text{ on}}$: 67.04 ± 10.17 s, $T_{1/2 \text{ off}}$: 970.95 ± 205.30 s) (Supplementary Fig. 9b, d). By contrast, a light-insensitive mutant of OptoFGFR1 (D387A in CRY2) did not affect Ras activity (Supplementary Fig. 9c). In addition, local delivery of light to a small subcellular region led to spatially restricted activation of Ras followed by membrane protrusion in the illuminated area (Supplementary Fig. 9e, f); repeated and local activation of FGFR induced cell migration to the blue light along the activity gradient of Ras (Supplementary Fig. 9g and Supplementary Movie 4). Next, we monitored HRas activity in cultured rat hippocampal neurons under light-mediated activation of TrkB signaling using OptoTrkB[23] (Fig. 2a). We co-expressed OptoTrkB and R-HRas in hippocampal neurons (DIV-9−12). Again, under global and transient activation of OptoTrkB, fluorescence intensity of R-HRas was augmented ~ 1.5-fold and saturated within five minutes (Fig. 2b, c). Light illumination on a small dendritic region induced reversible HRas activation specifically in stimulated dendrites (Fig. 2d, e Supplementary Fig. 10 and Supplementary Movie 5). Also, under local and persistent delivery of light on the peripheral of cell body, we could observe substantial increase of Ras activity near the illumination site accompanied by membrane protrusion (Supplementary Fig. 11a). Interestingly, we found that proximal dendrites showed growth of filopodia-like structures where Ras activity was selectively detected at the base rather than within the filopodia (Supplementary Fig. 11b, c), indicating spatially confined regulation of Ras activity by TrkB during this process of morphological change.

Next, we tried to locally activate OptoTrkB at distal parts of neurites of cultured hippocampal neurons at early stage of differentiation (DIV-2) in which neurites have not been specified as axons or dendrites yet. Under illumination of light on a distal part of neurite, we found spatially restricted activation of Ras and efficient elongation of the neurite. In contrast, non-illuminated neurites did not show any extension or increased Ras activity; rather, they were retracted along with attenuation of Ras activity (Supplementary Fig. 12a). When the stimulation site was changed to other neurites, we observed the same effects of shifted balance of Ras activity and selective elongation of neurites. In addition, when we illuminated light on cell body, the previously extended neurite dramatically shrank. Thus, these results indicate that during the process of selective neurite outgrowth, subcellular regions undergo a competition to achieve neuronal polarization probably through utility of limited pools of key molecules such as Ras or actin[24,25].

As the previous study reported that local activation of Ras has an important role in axon formation[24], we analyzed kinetic correlations between Ras activity and growth cone extension

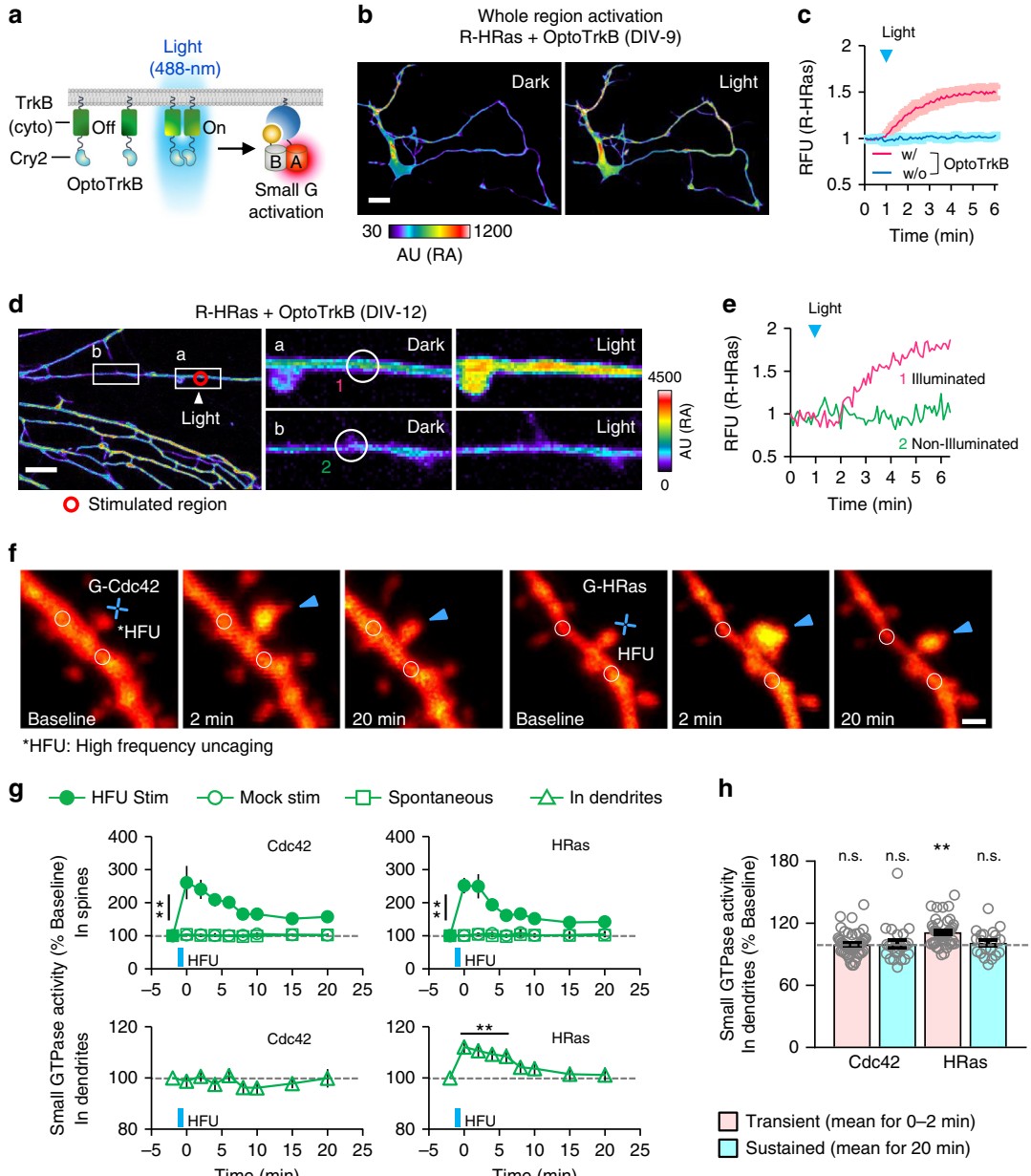

**Fig. 2** Spatiotemporal visualization of small GTPase activation in dendrites and single spines upon optoTrkB activation and LTP induction. **a** Schematic of Ras activation by blue light-mediated OptoTrkB activation. **b** Cultured hippocampal neuron (DIV-9) showing fluorescence change of R-HRas upon blue light-induced whole cell activation of OptoTrkB. Whole-cell activation; light was globally illuminated in the whole field of view. **c** Time-lapse measurement showing fluorescence change of R-HRas upon light stimulation. $n = 4$ (red), 6 (blue). Scale bar, 20 μm. **d** (left) images showing local fluorescence increase of R-HRas intensity upon OptoTrkB activation. Local stimulation was applied to a small region of dendrite (indicated by red circle) of hippocampal neuron (DIV-12) at $t = 1$ min. (right) Enlarged images of the indicated regions (white boxes) showing fluorescence change of R-HRas at the region of illumination and a distal dendrite. Scale bar, 20 μm. **e** Time-lapse graph showing fluorescence changes of R-HRas within regions indicated by white circles in **d** (magenta; nearby stimulation, green; distal from stimulation). **f** Time-lapse images of dendrites from layer 2/3 pyramidal neurons co-expressing tdTomato (red) and small GTPases sensors, G-Cdc42 (left) and G-HRas (right). Target spines (arrowheads) were exposed to HFU (blue crosses). **g** Time-lapse measurement showing fluorescence changes of G-Cdc42 and G-HRas in HFU-stimulated spines (top) and nearby dendrites (bottom, indicated by white circles in **f**, < 2.5 μm from target spines). Filled circles, G-Cdc42: **$P < 0.01$ at all post-HFU time points; $n = 26$ spines, 22 cells; G-HRas: **$P < 0.01$ at all post-HFU time points; $n = 22$ spines, 19 cells. Open circles, G-Cdc42: $n = 10$ spines, 10 cells; G-HRas: $n = 11$ spines, 11 cells. Open squares, G-Cdc42: $n = 23$ spines, six cells; G-HRas: $n = 23$ spines, six cells. Triangles (G-Cdc42), $n = 26$ spines, 22 cells. Triangles (G-HRas), **$P < 0.01$ up to 6 min; $n = 22$ spines, 19 cells. **h** Graph showing relative changes of fluorescence of G-Cdc42 and G-HRas in dendrites in different time periods (transient and sustained) following HFU. **$P < 0.01$. Statistical analysis were performed by Student's two-tailed $t$ test; n.s., not significant. *HFU: high-frequency uncaging; RFU: relative fluorescence unit; AU: arbitrary unit; Error bars: s.e.m. Images or quantified data are representatives of multiple experiments ($N > 3$)

under local stimulation of OptoTrkB. As a result, we found that activation of Ras was detected before initial response of neurite extension (Supplementary Figure 12b, c, d) and temporal cross-correlation analysis revealed that Ras activation preceded neurite extension by 0.534 min during the light stimulation (2–30 min) (Supplementary Fig. 12e). Altogether, we proved high compatibility of our red-shifted sensors with blue light-based optogenetic modules that allows us to achieve multiplexed analysis and investigate spatiotemporal roles of protein activity under light-driven perturbation on a certain biological process.

**Visualization of small GTPase activities in single spines under sLTP**. To test whether our small GTPase sensors are sensitive enough to visualize local signaling in complex multicellular environments, we examined individual dendritic spines in organotypic brain slices. We transfected G-Cdc42 or G-HRas together with tdTomato to visualize GTPase activity and neuronal morphology, respectively (Fig. 2f). Repetitive two-photon photolysis of glutamate (*HFU; high-frequency uncaging stimulus) on single dendritic spines increased dendritic spine volume without causing structural changes to neighboring dendritic spines (Supplementary Fig. 13a, b). Furthermore, G-Cdc42 fluorescence significantly increased at the target spines for 20 min (Fig. 2g, h and Supplementary Fig. 13c). These elevated signals were confined to the target spine head, with no fluorescence changes observed at a nearby dendritic region (2.5 μm away). Unlike Cdc42, increased Ras activity at the stimulated spine diffused to a nearby dendritic region, suggesting a broader spatial scale of Ras compared to Cdc42 activity[26].

**Monitoring Ras activity in the brains of freely behaving mice**. To test whether the G-HRas sensor effectively detects Ras activity in vivo, we injected adeno-associated viruses expressing G-HRas and tdTomato (to visualize cell morphology) into the primary motor cortex (M1) of mice (Fig. 3a). Individual neurons exhibited different levels of fluorescence intensities, expressed by a log-normal distribution curve when plotted by a G/R ratio (Fig. 3b, c). The coefficient of variation of G-HRas expression was significantly higher than that of tdTomato expression, suggesting that highly variable Ras activity in vivo did not originate from differential expression levels (Fig. 3d). A line scan measure on a neuronal cell body expressing G-HRas also confirmed a typical expression pattern of plasma membrane (Fig. 3e, f).

To examine whether G-HRas can monitor real-time changes of Ras activity in vivo, we applied brain-derived neurotrophic factor (BDNF) through a small cranial window. Time-lapse imaging of the same neuronal population showed a gradual increase in the average Ras activity upon BDNF (2 μg ml$^{-1}$) application (Fig. 3g, h) that became significantly enhanced compared with the control[16] (Fig. 3i). In contrast, the overall Ras activity was significantly reduced in an anesthetized state (Fig. 3j, k). Because neuronal activity in M1 decreases significantly in the anesthetized condition[27], we examined a putative correlation between Ras and neuronal activity. We injected R-HRas, a red Ras activity sensor, and a genetically encoded calcium indicator, GCaMP6s, into M1. We conducted in vivo two-photon imaging while head-fixed mice ran on a spherical treadmill (Fig. 3l, m). Post hoc analysis of time-lapse measures revealed a positive correlation between the average changes of GCaMP6s and R-HRas signals, suggesting that neurons with high activity tend to show high Ras activity (Fig. 3n, o).

**Visualization of individual Ras activity at single spine resolution**. To probe whether the sensor can report Ras activity at a single-synapse resolution in vivo, we imaged dendrites under high

magnification (Fig. 4a). We found that signals of Ras activity were distributed as discrete puncta throughout dendrites and ~20% of puncta showed high G-Ras intensity (>2.5-fold) (Fig. 4b). We monitored real-time changes of Ras activity while mice were behaving on the treadmill and found that individual punctum showed variable initial levels and changes of Ras activity (Fig. 4c, d), possibly owing to dynamic synaptic inputs with different strengths. To further characterize the dynamics of Ras activity, we categorized puncta based on their changes in Ras activity (same, > 80% and < 120%; up, ≥ 120%; down ≤ 80%) over 30 min. We found a significant increase in the puncta population in the "up" category, a substantial decrease in the "down" category, and no difference in the "same" category in the awake versus anesthetized state (Fig. 4e). Interestingly, puncta showing high versus low fluorescence intensity were more stable; a similar tendency did not exist in the anesthetized state (Fig. 4f). Next, we performed the same experiments using G-Rac1 and G-Cdc42 sensors. Even though overall fluorescence of G-Rac1 and G-Cdc42 was dimmer than G-HRas, we obtained consistent result in which puncta population of upregulation of small GTPase increased and puncta population of downregulation of small GTPase decreased while mice were running on the treadmill (Supplementary Fig. 14). These results demonstrate that our small GTPase sensors can visualize cellular signaling at a micron scale in awake-behaving animals.

## Discussion

In this study, we describe intensiometric biosensors that can visualize Ras and Rho small GTPase activities with a high spatiotemporal resolution. Using spectrally separated ddFP modules (GA and RA), we achieved simultaneous monitoring and direct comparison of multiple small GTPase activities in the same subcellular environment. Although a previous study showed that copy B could interchangeably interact with both GA and RA owing to their intrinsic binding properties[14], we could not detect noticeable signal cross-reactivity between Ras and Rho sensors, likely owing to the overwhelming binding affinity of small GTPases and effector proteins. Therefore, our experimental multiplexing approach will be valuable for reliable assessment of spatiotemporal coordination of multiple small GTPases in a given cellular context. Moreover, our results revealed a correlation between Ras function and neuronal activity represented by Ca$^{2+}$ concentration (Fig. 3n, o); thus, our sensors could be readily combined with other conventional biosensors to provide an integrative strategy for analyzing many signaling events related to small GTPases. We also demonstrated that red-shifted sensors are highly compatible with blue light-controllable optogenetic modules. This combination will enable the investigation of dynamic changes in protein activities under space- and time-resolved signaling perturbation. In parallel, red or near infrared light-based modulating systems[9,28,29] would be essentially compatible with both RA and GA sensors, allowing multiplexed visualization under optogenetic manipulation. Notably, our work demonstrated that the high sensitivity and signal-to-noise ratio of ddFP-based sensors enables small GTPase activity to be visualized in the intact brain of freely behaving animals at a micron scale. This suggests potential value of our sensor to uncover specific roles of small GTPases in various brain functions such as learning and memory under physiological conditions. Beyond studies in the brain, our sensors will be generally applicable to in vivo imaging of other tissues in normal and disease states. The development of sensors for other GTPase members and the expansion of the ddFP palette[30], accompanied by computational multiplexing[8], will greatly increase our ability to analyze complex signaling networks in diverse biological systems.

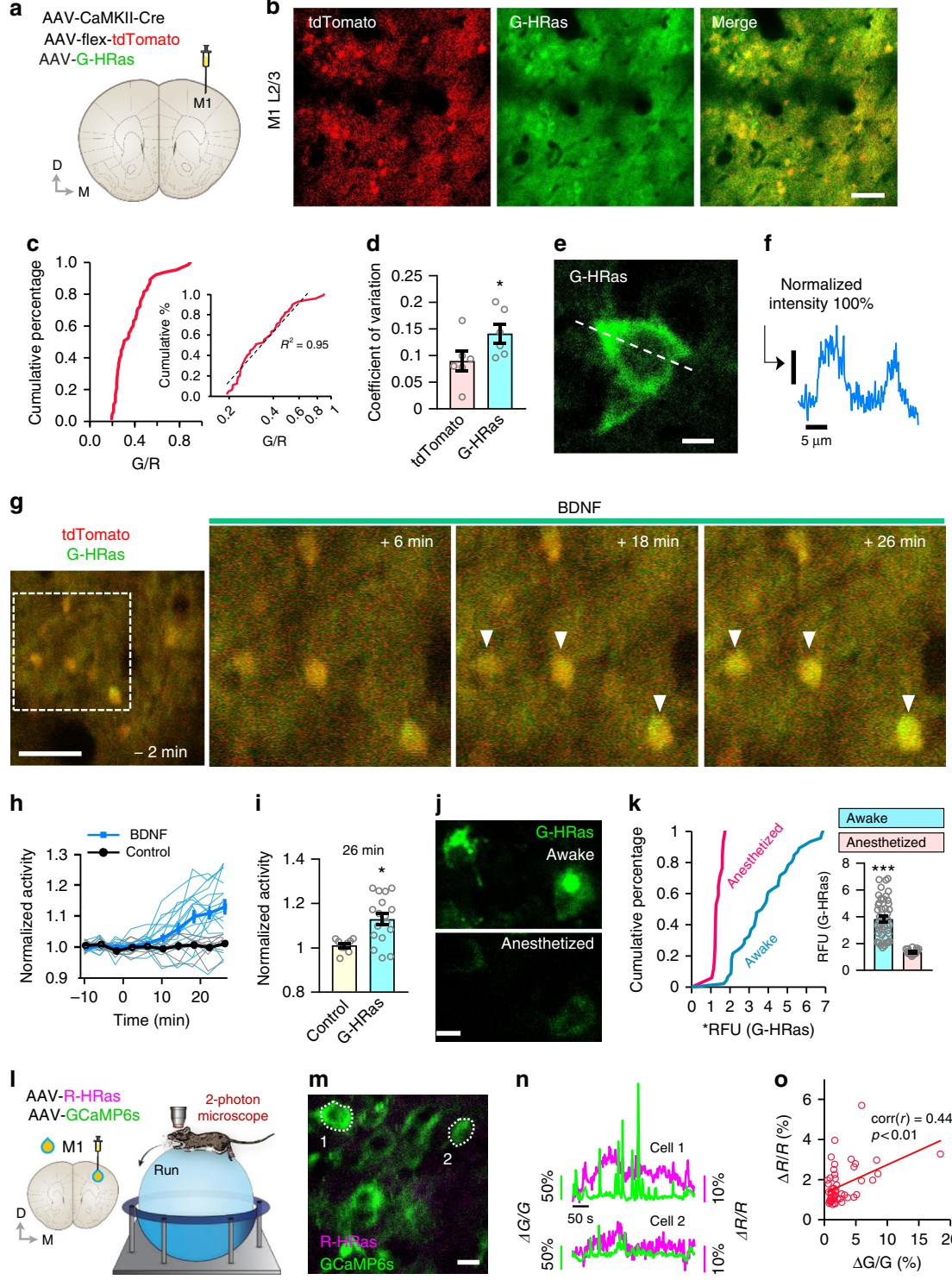

## Methods

**Plasmid construction**. Expression plasmids for pDDRFPA1-CaM, pM13-DDRFPB1(ref. [12]), pDDGFP-A, pDDGFP-B[13], (Addgene plasmid #36292, #36293, #40286, #40287, donated by Robert E. Campbell, University of Alberta, Canada), piRFP682-N1(ref. [17]) and pBAD/His-miRFP670(ref. [31]) (Addgene plasmid #45459, Addgene plasmid #79984, donated by Vladislav Verkhusha, Albert Einstein College of Medicine, USA), and ERKKTR-mClover[21] (Addgene plasmid #59150, donated by Markus Covert) were obtained from Addgene. EeeVee-Raichu-Rac1(ref. [15]) was donated by Michiyuki Matsuda, Kyoto University, Japan). In order to generate plasmids encoding pGA-C1, pRA-C1, pB1-C1, pB3-C1, the PCR-amplified sequences encoding ddFP indicated as GA (ddGFP-A), RA (ddRFP-B), B1 (ddGFP-B), or B3 (ddRFP-B), were cloned into pEGFP-C1 (Clontech), respectively,

after excising the original EGFP by *Age*I and *Bsr*GI. Expression plasmids for GA-KRas (S17N), GA-KRas (Q61L), B1-KRas (S17N), B1-KRas (Q61L), GA-KRas (wild-type), GA-HRas (wild-type), GA-NRas (wild-type), RA-KRas (wild-type), and RA-HRas (wild-type) were generated by inserting each mutant or wild-type K, H, N –Ras into *Bsr*GI and *Bam*HI sites of GA-C1, RA-C1, or B3-C1 vectors, respectively. To generate B1-RBD$_{Raf1}$ and GA-RBD$_{Raf1}$, RBD$_{Raf1}$ (amino acids 51–131) were inserted into *Xho*I and *Bam*HI sites of B3-C1 and GA-C1 vectors, respectively. For the generation of GA-GA and B3-B3 expression vectors, PCR-amplified GA and B3 were inserted into *Bsr*GI site of GA-C1 and B3-C1, respectively, using In-Fusion cloning system (Clontech) according to the manufacturer's instructions. In order to generate GA-GA-KRas (Q61L) and GA-GA-KRas (S17N), KRas (Q61L) or KRas (S17N) were inserted into *Bsr*GI and *Bam*HI sites of

**Fig. 3** Visualizing real-time dynamic changes of Ras activity in the intact brains of freely behaving mice. **a** Schematic depiction of virus injection and imaging positions. **b** In vivo two-photon imaging of G-HRas in layer 2/3 neurons. Scale bar, 100 μm. **c** Cumulative distribution of G/R ratio ($n = 65$ cells). Log-normal scaled plot of the distribution (inset, $R^2 = 0.95$). **d** Coefficient of variation for tdTomato and G-HRas signals (average ± s.e.m: 9.01 ± 1.89 for tdTomato; 14.2 ± 1.79 for G-HRas). **e** A representative image of G-HRas expression in the cell body. Scale bar, 10 μm. **f** Normalized intensity profile of G-HRas expression across the cell body, indicated as a dotted line in **e**. **g** Time-lapse imaging of G-HRas fluorescence with BDNF treatment. **h** Individual and average time course graph of normalized activity of G-HRas ($n = 17$ cells for BDNF; $n = 9$ cells for control). **i** Comparison of G-HRas activity between control and BDNF groups (average ± s.e.m: 1.01 ± 0.01 for control; 1.13 ± 0.025 for BDNF). **j** Representative images of G-HRas activity in awake (top) and anesthetized (bottom) states. **k** Cumulative distribution of normalized G-HRas intensity in awake and anesthetized states ($n = 46$ cells, awake; $n = 17$ cells, anesthetized). Comparison of average G-HRas intensity (inset) (average ± s.e.m: 3.84 ± 0.22, awake; 1.37 ± 0.05, anesthetized). **l** Schematics of virus injection and two-photon microscopy in mice running on a spherical treadmill. **m** A representative image of neurons expressing R-HRas and GCaMP6s. **n** Traces of R-HRas and GCaMP6s signals from two exemplary neurons. **o** Correlation between R-HRas and GCaMP6s signals ($n = 49$ cells, Pearson's $r = 0.44$). *$P < 0.05$, ***$P < 0.001$ by Student's two-tailed $t$-test. RFU: relative fluorescence unit. Images or quantified data are representatives of multiple experiments ($N > 3$). BDNF brain-derived neurotrophic factor

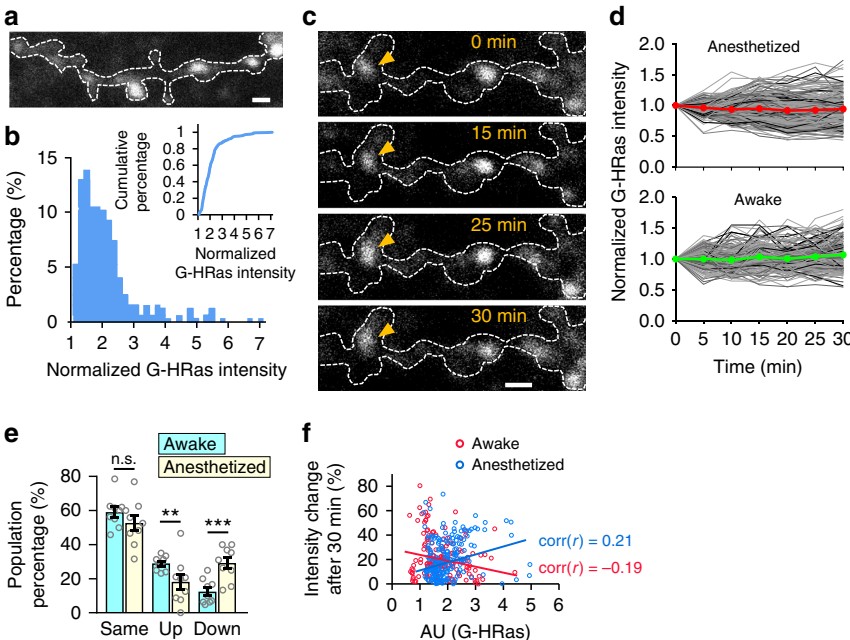

**Fig. 4** Visualization of dynamic Ras activity at the synapse resolution in awake mice. **a** Schematic representative image of G-HRas expression in a dendrite in vivo. **b** Distribution of G-HRas intensity ($n = 325$). Cumulative distribution (inset). **c** Time-lapse in vivo two-photon imaging of G-HRas activity in the awake state. Arrow indicates example Ras signal punctum that shows fluorescence intensity increases. **d** Time courses of activity in anesthetized and awake states. Gray scaled and colored (red, green) lines indicate changes of individual puncta and average, respectively ($n = 194$ for the anesthetized, $n = 165$ for the awake). **e** Categorized G-HRas responses at $t = 30$ min: same (>80 and <120%; average ± s.e.m: 59.07 ± 3.09 for the awake, average ± s.e. m: 52.72 ± 4.42 for the anesthetized), up (≥120%; average ± s.e.m: 28.89 ± 1.40 for the awake, average ± s.e.m: 18.19 ± 4.50 for the anesthetized), and down (≤80%; average ± s.e.m: 12.38 ± 2.39 for the awake, average ± s.e.m: 29.24 ± 3.33 for the anesthetized). **f** Correlations between G-HRas intensity and intensity change after 30 min for awake and anesthetized states (Pearson's $r = -0.19$ for the awake, $r = 0.21$ for the anesthetized). *$P < 0.05$, ***$P < 0.001$ by Student's two-tailed $t$ test, n.s., not significant. Error bars, s.e.m. All scale bars, 2 μm. Images or quantified data are representatives of multiple experiments ($N > 3$)

GA-GA-C1, respectively. In order to generate B3-B3- RBD$_{Raf1}$, PCR-amplified RBD$_{Raf1}$ was fused into *Bsr*GI site of B3-B3-C1 using In-Fusion cloning system (Clontech) according to the manufacturer's instructions. In order to generate RBD$_{Raf1}$-B3 and RBD$_{Raf1}$-GA, RBD$_{Raf1}$ were cloned into the B3-C1 and GA-C1 plasmids between *Nhe*I and *Age*I sites. To generate B3-RBD$_{Raf1}$-B3, PCR-amplified B3 was inserted into site of *Bam*HI at the C-terminus of B3-RBD$_{Raf1}$ using In-Fusion cloning system (Clontech) according to the manufacturer's instructions. iRFP682-conjugated GA-KRas (S17N), GA-KRas (Q61L), GA-Rac1 (T17N), GA-Rac1 (Q61L), GA-Cdc42 (T17N), GA-Cdc42 (Q61L) were generated by inserting KRas (S17N), KRas (Q61L), Rac1 (T17N), Rac1 (Q61L), Cdc42 (T17N), Cdc42 (Q61L), respectively, into *Bsr*GI and *Bam*HI sites of iRFP-GA-C1, after PCR-amplified iRFP682 was cloned into GA-C1 plasmid between *Nhe*I and *Age*I using In-Fusion cloning system (Clontech) according to the manufacturer's instructions. In order to generate RA-Rac1 (wild-type), RA-Cdc42 (wild-type), Rac1 (wild-type), and Cdc42 (wild-type) were inserted into *Eco*RI and *Bam*HI sites of RA-C1 vector, respectively. In order to generate B3-CRIB$_{Pak1}$, PCR-amplified CRIB$_{Pak1}$ (amino acids 69–108) was inserted into *Bsr*GI and *Bam*HI or *Eco*RI and *Bam*HI sites of B3-C1 vector, respectively. To create bicistronic expression vectors for the green or red KRas, HRas, and NRas sensors under CMV promoter, the sequences of B3-RBD$_{Raf1}$ and GA-KRas, RA-KRas, GA-HRas, RA-HRas, or GA-NRas were

PCR-amplified and fused to EGFP-C1 (Clontech) vector after excision of EGFP at the site of *Nhe*I and *Bam*HI. Complement 2A sequences, as previously described[18], were flanked both at the C-terminus of B3-RBD$_{Raf1}$ and N terminus of GA or RA-fused Ras sequences for In-Fusion reaction. For generation of bicistronic expression vectors of the red Rac1 and Cdc42 sensors, the sequences of B3-CRIB$_{Pak1}$ and RA-fused Rac1 or Cdc42 were PCR-amplified and fused to EGFP-C1 (Clontech) vector after excision of EGFP at the site of *Nhe*I and *Bam*HI. Complement 2A sequences[18], were flanked both at the C terminus of B3-CRIB$_{Pak1}$ or B3-CRIB$_{Wasp}$ and N terminus of RA-fused Rac1 and Cdc42 sequences for In-Fusion cloning reaction according to the manufacturer's instructions. The myristoylation sequence was added to the N terminus of iRFP682-C1 or miRFP-C1 to generate Lyn-iRFP682 or Lyn-miRFP, using the In-Fusion cloning system (Clontech) according to the manufacturer's instructions. The Lyn-FRB and FKBP-SOS1, FKBP-iSH2, FKBP-Tiam1, FKBP-Vav2 were generated as previously described[20]. Plasmids for Lyn-cytFGFR1-CRY2-mCit were generated as previously described[22]. To create pCAG-driven bicistronic expression vectors for G-HRas, R-HRas, and G-Cdc42, B-RBD$_{Raf1}$ with GA-HRas or RA-HRas, and B-CRIB$_{Pak1}$ with GA-Cdc42 were PCR-amplified and inserted into *Xho*I and *Not*I sites of pCAG-IRES-EGFP-C1 vector using Gibson assembly (NEB) according to the manufacturer's instructions. In order to generate AAV viral expression vectors

for CAG-G-HRas and R-HRas, B-RBD$_{Raf1}$ with GA-HRas or RA-HRas from pCAG-G-HRas and pCAG-R-HRas were PCR-amplified and inserted into *Xba*I and *Eco*RV sites of pAAV-CAG-FLEX-EGFP (Addgene plasmid, #28304) plasmid, respectively, using Gibson assembly (NEB) according to the manufacturer's instructions. In order to generate pAAV-CaMKIIa-Lyn-cytTrkB-PHR-HA-WPRE viral expression vector, Lyn-cytTrkB-PHR was PCR-amplified, flanked by *Kpn*I and *Eco*RI, and ligated into pAAV-CK (0.4) GW (Addgene plasmid #27226). Lyn-cytTrkB-PHR-EGFP was PCR-amplified, flanked by *Kpn*I and *Eco*RI, and ligated into pAAV-CK (0.4) GW (Addgene plasmid #27226) in order to generate pAAV-CaMKIIa-Lyn-cytTrkB-PHR-EGFP-WPRE viral expression vector.

**Cell culture and transfection.** HeLa and MDA-MB-231 (ATCC) cells were maintained in Dulbecco's modified Eagle's medium (PAA Laboratories GmbH) supplemented with 10% fetal bovine serum (FBS; Invitrogen) at 37 °C and 10% $CO_2$. We have not recently authenticated these cell lines, but we have confirmed that they are free from mycoplasma contamination (tested with e-MycoTM Mycoplasma PCR detection kit (ver. 2.0), iNtRON). Cells were transfected using a Microporator (Neon Transfection System, Invitrogen) or Lipofectamine LTX (Invitrogen) according to the manufacturer's instructions, except that conditions of electroporation were optimized to increase efficiency of transfection. The optimized conditions of electroporation for HeLa or MDA-MB-231 cells were two pulses of 980 V for 35 ms or two pulses of 950 V for 30 ms, respectively. For live-cell imaging, HeLa cells were plated in a micro plate 96-well ibiTreat (ibidi). MDA-MB-231 cells were seeded on a 96-well glass bottom plate coated with 5 μg ml$^{-1}$ (fibronectin) (Sigma).

**Preparation and transfection of hippocampal neurons.** E18 pregnant Sprague–Dawley female rats were prepared for hippocampal cultures. All experimental procedures were approved by the Animal Ethic Committee at the Korea Advanced Institute of Science and Technology (KAIST; Daejeon, Korea). In brief, embryos were obtained from the rats and then placed into Hank's Balanced Salt Solution (HBSS) (cat. no. 14185-052; Gibco)–HEPES (10 mM; cat. no. 15630-080; Gibco) solution. Hippocampi were dissected from the embryos and incubated in 0.25% trypsin for 25 min at 37 °C with tapping every 5 min. Hippocampi were washed for three times with HBSS–HEPES, and triturated with a fire-polished Pasteur pipette. Tissues were dissociated after 10–20 trituration, and neurons were immediately plated in pre-equilibrated dishes or plates coated with 1 mgml$^{-1}$ poly-L-lysine (cat. no. P2636; Sigma) in plating medium. The plating medium consisted of Neurobasal medium (cat. no. 21103-049; Gibco) supplemented with 2% FBS, 2% B-27 (cat. no. 17504-044; Gibco), 2% glutamax (cat. no. 35050-061; Gibco), and 2% penicillin–streptomycin (cat. no. 15140-122; Gibco). The neurons were incubated at 37 °C in a humidified 5% $CO_2$ atmosphere. Plating medium were replaced with maintaining medium (plating medium without FBS) within 6–24 h. Neurons were transfected with AAV-mediated gene delivery.

**Live-cell imaging and electronics.** For imaging Live-cell imaging was performed using a Nikon A1R confocal microscopy (Nikon Instruments) mounted onto a Nikon Eclipse Ti body equipped with CFI Plan Apochromat VC objectives (×60, 1.4 numerical aperture; Nikon) along with digital zooming of Nikon imaging software (NIS-elements AR 64-bit version 3.21, Laboratory Imaging). Chamlide TC system placed in a microscope stage was used to maintain environmental condition at 37 °C and 10% $CO_2$ (Live Cell Instrument, Inc.). For treatment of the chemical stimulus, HeLa cells were serum starved 3 h prior to imaging, and the medium was replaced with DPBS (Invitrogen) just before the imaging.

**Blue light-excitation of optogenetics modules.** Whole-cell photoexcitation to induce optoFGFR1 and optoTrkB was delivered using a photostimulation module in Nikon imaging software (NIS-elements) that provided three loops of 0.5 s stimuli. A laser power of 221 μW mm$^{-2}$ (measured with an optical power meter from ADCMT) was used for photoexcitation. In case of local photoexcitation, three pulses in 0.189 s with 315 mW mm$^{-2}$ was delivered for optoFGFR1 activation. Repeated illumination (53 mW mm$^{-2}$, three pulses in 0.189 s at 1 min intervals) was applied to the peripheral region of an MDA-MB-231 cell for directed migration. Two pulses in 0.126 s with 412 mW mm$^{-2}$ was delivered for local optoTrkB activation in hippocampal culture neuron.

**Image processing and analysis.** Images were analyzed with Nikon imaging software (NIS-elements AR 64-bit version 3.21, Laboratory Imaging) and Meta-Morph software (version 7.8.12, MDS Analytical Technologies). For quantification of dynamic range the sensors, fluorescence intensity (500 ~ 2500 arbitrary units) at designated regions of interest were measured using the "Annotations and Measurements" tool in Nikon imaging software. For cell area analysis, the "Automated Measurements" tool and "ROI Statistics" tool in Nikon imaging software were used. In order to quantitatively analyze the changes of fluorescent intensities during activation of each sensors, the "Time Measurement" tool in Nikon imaging software was used. Images were denoised in "2D Current Plane" and "2D + time" using Safir software (version 1.0.3, Roper Scientific)[32], and running into Meta-Morph software. A "Kymograph" tool in MetaMorph software was used to draw kymographs. For protrusion and retraction analysis, fluorescence images of Lyn-

iRFP682 were converted into binary images based on intensity thresholding with MetaMorph Software. Using the "Arithmetic" tool, images of retraction were obtained by subtracting the binary images of before and after light stimulation. Regions of protrusions were conversely isolated. Areas that overlapped right before stimulation were obtained by operating "Logical AND" with the two binary images. Isolated regions were combined with the "Color Combine" tool. Statistical significance was evaluated by Student's two-tailed *t* test.

**Preparation and transfection of organotypic cortical slice cultures.** Organotypic slice cultures from mouse somatosensory cortex were prepared from C57BL/6 wild-type mice (Charles River Laboratory) at postnatal day 2 (P2)-P3, as described previously[33], in accordance with the Institutional Animal Care and Use of Max Planck Florida Institute for Neuroscience and National Institutes of Health guidelines. Slices were transfected 7–10 days prior to imaging using biolistic gene transfer[34]. A total of 10 μg of tdTomato[33] and 20 μg of pCAG-B3-CRIB$_{Pak1}$-2A-GA-Cdc42 or 10 μg of tdTomato and 20 μg of pCAG-B3-RBD$_{Raf1}$-2A-GA-HRas were coated onto 6–7 mg of gold particles. The age of the culture is reported as equivalent postnatal (EP) day; postnatal day at slice culturing + days *in vitro*.

**Time-lapse two-photon imaging and uncaging.** Two-photon imaging and uncaging were performed at EP 15–21 on transfected layer 2/3 pyramidal neurons within 40 μm of the slice surface at 30 °C in recirculating artificial cerebrospinal fluid (ACSF; in mM: 127 NaCl, 25 NaHCO$_3$, 1.25 NaH$_2$PO$_4$, 2.5 KCl, 25 D-glucose, aerated with 95%O$_2$/5%CO$_2$, ~ 310 mOsm) with 2 mM CaCl$_2$, 0 mM MgCl$_2$, 2.5 mM 4-methoxy-7-nitroindolinyl-caged-L-glutamate (MNI-glutamate), and 0.001 mM tetrodotoxin. For each neuron, image stacks (512 × 512 pixels; 0.035 μm/pixel) with 1 μm z-steps were collected from one segment of secondary or tertiary apical dendrites at 2 or 5 min intervals using a two-photon microscope (Bruker, Inc) with a pulsed Ti::sapphire laser (MaiTai HP DeepSee, Spectra-Physics) tuned to 920 nm (3–4 mW at the sample). All images shown are maximum projections of 3D image stacks after applying a median filter (2 × 2) to the raw image data. Uncaging of MNI-glutamate was achieved as described[33]. In brief, LTP-inducing HFU consisted of 30 pulses (720 nm; 15–18 mW at the sample) of 5 ms duration delivered at 1 Hz in the presence of 2.5 mM MNI-glutamate by parking the beam at a point ~ 0.5 μm from the center of the spine head with a pulsed Ti::sapphire laser (MaiTai HP, Spectra-Physics). The mock stimulus was identical in parameters to the HFU stimulus, except carried out in the absence of MNI-glutamate. No more than two LTP trials were performed from the same neuron.

**Image analysis and quantification of glutamate uncaging.** Estimated spine volume and G-Cdc42 or G-HRas enrichment on dendritic spines were measured in fluorescence images from red (tdTomato) and green (Cdc42 or HRas) channels using ImageJ (NIH). Integrated fluorescence intensities were calculated from background-subtracted and bleed-through-corrected red and green fluorescence using the integrated pixel intensity of a boxed region surrounding the spine head, as described previously[34]. G-Cdc42 or G-HRas fluorescence intensities in dendritic shafts were calculated from background-subtracted and bleed-through-corrected green fluorescence intensities (as described above) by examining two regions of interest (~ 0.5–1 μm$^2$) on dendrites located < 2.5 μm from the nearby HFU-stimulated spines.

**SNR analysis.** Having multiple standard points, we used following equation to determine qualitative detection limit[35]:

$$y_D = y_0 + Ps + Qs \tag{1}$$

Where $y_D$ is signal at detection limit concentration $x_D$, $y_0$ is estimate of response signal for a blank, $s$ is estimate of residual variance. $P$ and $Q$ are defined as:

$$P = t_{1-\alpha}\sqrt{\frac{1}{n} + \frac{1}{N} + \frac{\left(\bar{\lambda} + \frac{1}{R}\right)^2}{\sum\left(\lambda_i - \bar{\lambda}\right)^2}} \tag{2}$$

$$Q = t_{1-\beta}\sqrt{\frac{1}{n} + \frac{1}{N}} \tag{3}$$

Where $t_{1-\alpha}$, $t_{1-\beta}$ are the Student's $t$ corresponding to $N$-2 degrees of freedom for $(1-\alpha)$ and $(1-\beta)$ or evading error of the first kind and second kind, respectively.[36] $N$ is the number of standards, and $n$ is the number of measurements. $Q$ holds if content of standard fall in the neighborhood of detection limit. For $R$ and $\lambda$,

$$R = \frac{(x_N - x_1)}{x_1} \tag{4}$$

$$\lambda_i = \frac{x_i - x_1}{x_N - x_1} \tag{5}$$

Where $x_i$ is the concentration of the element of the interest in the $i$ th standard and $\lambda_i$ is the mean of $\lambda_i$'s. As we calculated change in the intensity of fluorescence

by subtracting basal signals, thus $y_0$ is 0. Background variance $\sigma$ can replace the residual variance $s$ if sample is large. Therefore, we can conclude that $y_D$ is expressed in the multiples of $s$, which is the SNR ratio[37]: Thus, equation (1) is now

$$y_D = (P + Q)\sigma \qquad (6)$$

As equation (1) assumes that standards have straight line, we introduced a log concentration $X_i = 3 + \log_{10}(2 \times x_i)$ to calculate detection limit with SNR ratio from standards, which have an exponential correlation. With our standard condition, $P$ and $Q$ only depends on relative ratio $\lambda$ therefore changes in constants of $X_i$ does not affect the value of $P$ and $Q$. Also, given that $\sigma$ stays similar between experiments and thereby considered as constant, we can use the value of $P+Q$ as threshold of detection limit in SNR ratio value, which is multiplies of $s$ for regression. Now the equation (6) can be rewritten in SNR ratio, as follows:

$$Y_D = P + Q \qquad (7)$$

As log of 0 cannot be calculated, we excluded result of 0 and used 0.005 to 100, which corresponds to 1 and 5.301 in $X_i$. With given standard condition, $R$ is 4.301, $\bar{\lambda}$ is 0.554, $N$ is 6 and the number of measurements is 10 at least. When $\alpha = \beta = 0.02$ and the degree of freedom is 4, $t = 2.999$. Thus the value of $P+Q$ is 4.446.

The standard result is calculated with a regression curve;

$$SNR\ value = k_{sensor} \times X_i \qquad (8)$$

When $\alpha = \beta = 0.02$, the threshold SNR value is 4.446, the detection limit $x_D$ for our ddFP biosensor is 0.011 ng ml$^{-1}$ and 24.946 ng ml$^{-1}$ for FRET sensor (see Supplementary Fig. 5)

**Animal surgery and stereotactic viral injection.** C57BL/6 wild-type mice (4–9 weeks, either gender, Jackson laboratory, Bar Harbor, ME, USA) were used for in vivo experiments. in accordance with protocols approved by the Max Planck Florida Institute for Neuroscience Institutional Animal Care and Use Committee and National Institutes of Health guidelines. Surgeries were performed on 4–6-week-old mice. Mice were anaesthetized with an intraperitoneal injection of an anesthetic cocktail containing ketamine (80 mg/kg) and xylazine (12.5 mg/kg) (Sigma-Aldrich). Hair on a surgical area was removed with a hair remover lotion (Nair, Church & Dwight Co, Inc; Princeton, NJ, USA) and ophthalmic ointment (Puralube Vet Ophthalmic Ointment) was applied to prevent eyes from drying. Next, the animal was placed in a stereotaxic device (Kopf, Model 900 Small Animal Stereotaxic Instrument). The surgical area was scrubbed by 10% betadine solution (Purdue product LP, Stamford, CT, USA) and cleaned. Body temperature (37–38 °C) was maintained by a thermostatically controlled heating pad (Harvard Apparatus, Holliston, MA, USA). Small incision was made on the scalp. Small craniotomy (~ 0.5 mm in diameter) was made above the injection site (the right motor cortex, A/P: +1.5 mm, M/L: +1.2 mm from the bregma, D/V: −0.25 mm from the brain surface). A mixture of a mixture of AAV1.DJ/8.CAG.G-HRas (600 nl), AAV1.CaMKII0.4.Cre.SV40 (200 nl), and AAV1.CAG.Flex.tdTomato.WPRE.bGH (200 nl) for population imaging, or a mixture of AAV1.hSyn.GCaMP6s.WPRE.SV40 (200 nl, Penn Vector Core) and AAV1.DJ/8.CAG.R-HRas (400 nl) for calcium imaging, or a mixture of AAV1.DJ/8.CAG.G-HRas (400 nl) and AAV1.CAG.tdtomato.WPRE.SV40 (200 nl, 1:100 diluted) for dendrite imaging were used for virus injection. The viral constructs were injected via a bevelled glass micropipette (tip size 10–20 μm diameter, Braubrand) backfilled with mineral oil. Flow rate (100 nl/min) was regulated by a syringe pump (World Precision Instruments). Following virus injection, skin adhesive (3M vetbond) was applied to close the incision site. General analgesia (Buprenorphine SR, 0.05 mg/kg) was injected subcutaneously and mice were monitored until they recovered from anesthesia. After around four weeks later, mice were anesthetized and hair was removed. A scalp was removed in a circular shape, and the surface of skull was cleaned. Cranial window (3 mm diameter) was implemented on the virus injection site and a custom-made headplate was attached to the exposed skull with the dental adhesive (C&B Metabond, Parkell inc, Edgewood, NY, USA).

**In vivo two-photon imaging.** Mice were head-restrained on the air-supported spherical treadmill. For mice with the cranial window, imaging was conducted through the cranial window with a two-photon microscope (Bruker) with excitation at wavelengths of 920 nm (MaiTai HP DeepSee, Newport Spectra-Physics) and 1045 nm (HighQ-2, Newport Spectra-Physics) while they were freely moving on the treadmill. For mice with BDNF treatment and control groups, imaging was conducted through a small open-skull window (3 mm diameter). For the anesthetized group, mice were anaesthetized with an intraperitoneal injection of an anesthetic cocktail containing ketamine (80 mg/kg) and xylazine (12.5 mg/kg) (Sigma-Aldrich) 10 min prior to imaging. Layer 2/3 (100 ~ 300 μm deep from the surface) of motor cortex was imaged. Imaging data set was motion-corrected using a custom-written MATLAB code based on full-frame cross-correlation image alignment algorithm. Regions of interests (ROIs) were semi-manually drawn based on fluorescence intensity, size, and shape by visually inspecting movies. For calcium imaging data set, all pixels within each ROI were averaged to create a fluorescence time series, F. The time-varying baseline (F0) of a fluorescence trace was estimated by the following procedure. Preliminary baseline Fluorescence time series, Pre F0, were loess smoothed with 120 frames. Preliminary ΔF, Pre ΔF, was obtained by subtracting Pre

F0 from F. Noise of Pre ΔF was estimated by subtracting loess smoothed Pre ΔF from the standard deviation of Pre ΔF. Offset of ΔF was determined by the mean of the distribution of Pre ΔF that does not exceed two times of noise. Baseline fluorescence trace, F0, was estimated by adding offset to Pre F0. ΔF/F for neuronal ROIs was obtained by subtracting F0 from F and dividing it with F0.

**BDNF treatment.** Human recombinant BDNF was purchased from Tocris. After ~4 weeks from the virus injection, mice were anesthetized with isoflurane. A scalp was removed and the surface of skull was cleaned. A small craniotomy (3 mm diameter) was made on the virus injection site and a custom-made headplate was attached to the skull with the dental adhesive (C&B Metabond). Once they recovered from anesthesia, mice were put on the air-floating spherical treadmill and in vivo two-photon imaging was conducted with ACSF containing (in mM): 124 NaCl, 26 NaHCO$_3$, 10 glucose, 3 KCl, 1.25 NaH$_2$PO$_4$, 2.5 CaCl$_2$, and 1.3 MgSO$_4$. For time-lapse two-photon imaging with BDNF application, normal ACSF was replaced with ACSF containing BDNF (2 μg ml$^{-1}$).

**Reporting summary.** Further information on experimental design is available in the Nature Research Reporting Summary linked to this article.

## Data availability
Supporting data of this study are available from the corresponding author on reasonable request.

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

## Acknowledgements

We thank all members of the Heo Laboratory for their supports and advices. We thank H.W. Yang (Stanford) for fruitful comments on sensor development and analysis. We thank D. Lee for drawing the mouse ball maze. This work was supported by the Institute for Basic Science (no. IBS-R001-G1) (to W.D.H.); KAIST Institute for the BioCentury, Republic of Korea (to W.D.H.); the Max Planck Florida Institute for Neuroscience (to H-B.K.); the NIH (R01MH107460 to H-B.K.); and the NIH (DP1MH119428 to H-B.K).

## Author contributions

W.D.H., J.K., and S.L. conceived the idea and directed the work. J.K., S.L., K.J., W.C.O., H-B.K., and W.D.H. designed experiments; J.K., S.L., K.J., W.C.O, and S.S. performed experiments; N.K., and Y.J.J. discussed computational analysis; J.K., S.L., K.J., W.C.O., H-B.K., and W.D.H. wrote the manuscript.

## Additional information

**Competing interests:** The authors declare no competing interests.

