## [Peer Review File · Nature Communications]

Reviewers' comments:

Reviewer #1 (Remarks to the Author):

This manuscript from Heo and colleagues introduced a suite of single-wavelength intensimetric small GTPase activity indicators based on dimerization dependent fluorescent proteins. These newly developed sensors enabled simultaneous visualization of Ras and Rho activities in single cells and direct multiplexing of optogenetics actuation and GTPase activity reporting. The authors also demonstrated that these sensors can be used for monitoring of GTPase activities in single spines as well as in vivo in freely behaving mice. This work represents a substantial achievement in both sensor engineering and biological applications. The experiments are generally well-designed and carefully controlled. I thus recommend publication of this manuscript in Nature Communications, once the following concerns and questions have been addressed.

1. A brief explanation of why chose B3 over B1 would be helpful.
2. The authors mentioned that "G-KRas sensitively responded to even a pg ml⁻¹ concentration of EGF (Fig. 1e)". However, it is not convincing from the figure. It will be better if the authors could quantify signal to noise ratio (SNR) and define the detection limit (say SNR>3).
3. It would be more appropriate to compare SNR between the intensimetric and FRET sensor instead of intensity change of G-HRas versus ratio change of RaichuEV-HRas.
4. Acronyms like RFU and HFU should be defined with full names.
5. In Fig.4, why are Ras signal changes highly variable? What are the possible reasons that high intensity puncta show a smaller signal change in the awake state? The tendency difference between awake state and anesthetized state seems negligible.
6. In Supplementary Fig. 1, it would be better to show the actual iRFP fluorescence image instead of white dotted line representation.
7. For the experiments in Supp Fig.7, it would better to include another set of control with G-Rac1 and RA-HRas-IRES-Lyn-iRFP.
8. Analysis of variance (ANOVA) should be used when comparing the values from multiple samples, including Fig. 1df, Fig. 2h, Supp Fig. 3c, and Supp Fig. 10bc. Detailed explanations of statistical analyses should be better described. In addition, the use of s.e.m. or s.d. for error bar should be consistent.

Reviewer #2 (Remarks to the Author):

In this paper Kim and colleagues present a new brand set of genetically encoded probes to evaluate changes in the activity of small GTPases. The approach these authors choose combines the use of red-shifted based sensors with blue-light optogenic modules. Such strategy would allow to measure changes in the activity of small GTPases in a temporal and spatially manner. As proof of concepts authors presented probes developed to evaluate the activity of Ras, Rac1 and Cdc42. The paper is correctly written and results presented are sound. However, there are a number of issues that need to be addressed to deliver a more convincing message to readers.

- 1.- In abstract section authors refers to small GTPases as enzymes. This is not correct, please changes enzymes by proteins

2.- At the beginning of the results section, authors would quickly introduce what are the effector domains chosen for each GTPases, explicitly indicating why they choose these domains over others.

3.- The Supplementary Figure 3C indicate in first panel that upon expression of ISH2, there is a significant increase of R.HRas activity which contradicts authors statements in the manuscript as follow: "We also confirmed that some activators, such as iSH2 (activator of endogenous PI3K) and Vav2, could increase fluorescence signals of both RRac1 and R-Cdc42, but not R-HRas". Please correct or clarify

4.- Authors state that "Quantification of local signal distribution revealed that the localized activity pattern of R-HRas and G-Rac1 showed no significant difference in comparison with RaichuEV-HRas and RaichuEV-Rac1 respectively". If this is true, what would be the benefit of using G-Rac1 or R-HRas instead of Raichu probes?. It is clear for this reviewer there are some benefits, but the message conveyed by authors get blurred with this statement.

5.- I have some concerns regarding Figure 2b as follow. It is not clear where in neurons light-activation of TrkB was induced. It may be very helpful to describe more in depth the experimental setup to perform these experiments. Additional, figure 2b should indicate the site for light stimulation in neurons. Is it the same if stimulation is done in axon, cell body or other neurites? Were there some morphological changes in neurons induced after Ras local activity? Such information is very relevant because it is widely accepted that a local Ras activation precedes axon specification (Fivaz et al 2000)

6.- Please include HFU acronym spelling in the results section. This will enhance readability and comprehension by readers

7.- Regarding Figure 4a, I believe that results presented may support authors idea that these probes are useful for in vivo studies. However, it is not clear why authors choose to monitor Ras at the single synapse level instead of Rac or Cdc42 which are better proxys of the morphological changes dendritic spines undergoes upon neuronal activity. The paper would be very much strength if authors provide evidences for Rac and Cdc42 at synaptic sites.

Point-by-point responses to reviewers' comments

We thank all the reviewers for their constructive comments which were very helpful for us to identify critical issues and improve our manuscript. We have addressed all the reviewers' points and carried out the requested experiments. Newly added figures are summarized as a table below. And revised parts in the manuscript are designated in yellow.

Newly added figure	Title
Supplementary Fig. 3	Comparison of SNR(signal-to-noise ratio) between intensimetric and ratiometric Ras sensor.
Supplementary Fig. 8	Examination of cross-reactivity between RA-HRas and GA-Rac1 in a single cell.
Supplementary Fig. 11	Visualization of Ras activity and morphological changes of cultured hippocampal neurons under local activation of OptoTrkB.
Supplementary Fig. 12	Monitoring of Ras activity and neurite morphology in a cultured hippocampal neuron under local activation of OptoTrkB.
Supplementary Fig. 14	Visualization of Rac1 and Cdc42 activity at the synapse resolution in awake behaving mice.

Reviewer #1:

We are grateful to reviewer #1 for positive assessment of our work and the constructive comments. According to those comments, now we have made following changes and improved our manuscript.

Remarks to the Author:

Summary

This manuscript from Heo and colleagues introduced a suite of single-wavelength intensimetric small GTPase activity indicators based on dimerization dependent fluorescent proteins. These newly developed sensors enabled simultaneous visualization of Ras and Rho activities in single cells and direct multiplexing of optogenetics actuation and GTPase activity reporting. The authors also demonstrated that these sensors can be used for monitoring of GTPase activities in single spines as well as in vivo in freely behaving mice. This work represents a substantial achievement in both sensor engineering and biological applications. The experiments are generally well-designed and carefully controlled. I thus recommend publication of this manuscript in Nature Communications, once the following concerns and questions have been addressed.

Specific comments:

1. A brief explanation of why chose B3 over B1 would be helpful.

According to the reviewer's comment, we have further described the reason for choosing GA-B3 pair rather than GA-B1 pair in our revised manuscript (Page 4). Briefly, we intended to select an A-B pair of ddFPs with a minimal basal fluorescence. In Supplementary Fig. 1, we examined fluorescence intensity generated by inherent dimerization of ddFP pairs (GA-B1 vs GA-B3) and found that GA-B3 pair showed much less fluorescence than GA-B1 pair, owing to weaker binding affinity ($K_d = 3 \mu\text{M}$, $40 \mu\text{M}$ for GA-B1, GA-B3, respectively). As the generated fluorescence by A-B dimerization regardless of GTPase-effector bindings can act as an artificial signal and reduce the dynamic range of sensors, use of B3 for design of small GTPase sensors would be more helpful than B1 copy.

2. The authors mentioned that "G-KRas sensitively responded to even a pg ml^{-1} concentration of EGF (Fig. 1e)". However, it is not convincing from the figure. It will be better if the authors could quantify signal to noise ratio (SNR) and define the detection limit (say $\text{SNR} > 3$)

3. It would be more appropriate to compare SNR between the intensimetric and FRET sensor instead of intensity change of G-HRas versus ratio change of RaichuEV-HRas

According to the reviewer's comments (Comment #2 and #3), we have performed SNR quantification to define the detection limit of ddFP sensor for EGF stimulation and to compare the efficiency between our sensors and FRET sensors as described below.

Having multiple standard points, we used following equation to determine qualitative detection limit (Hubaux and Vos, 1970):

$$y_D = y_0 + Ps + Qs$$

Where y_D is a signal at detection limit concentration x_D , y_0 is an estimate of response signal for a blank, s is an estimate of residual variance. P and Q are defined as:

$$P = t_{1-\alpha} \sqrt{\frac{1}{n} + \frac{1}{N} + \frac{\left(\bar{\lambda} + \frac{1}{R}\right)^2}{\sum(\lambda_i - \bar{\lambda})^2}}$$

$$Q = t_{1-\beta} \sqrt{\frac{1}{n} + \frac{1}{N}}$$

Where $t_{1-\alpha}$, $t_{1-\beta}$ are the Student's t corresponding to $N-2$ degrees of freedom for $(1 - \alpha)$ and $(1 - \beta)$ or evading error of the first kind and second kind, respectively (Currie, 1999), N is the number of standards, and n is the number of measurements. Q holds if content of standard fall in the neighborhood of detection limit.

For R and λ ,

$$R = \frac{(x_N - x_1)}{x_1}$$

$$\lambda_i = \frac{x_i - x_1}{x_N - x_1}$$

Where x_i is the concentration of the element of the interest in the i th standard and λ_i is the mean of λ_i 's. As we calculated change in the intensity of fluorescence by subtracting basal signals, thus y_0 is 0. Background variance σ can replace the residual variance s if sample is large. Therefore, we can conclude that y_D is expressed in the multiples of s , which is the SNR ratio (Henkelman, 1985): $y_D = y_0 + Ps + Qs$ is now $y_D = (P + Q)\sigma$. As equation 1 assumes that standards have straight line, we introduced a log concentration $X_i = 3 + \log_{10}(2 \times x_i)$ to calculate detection limit with SNR ratio from standards, which have an exponential correlation. With our standard condition, P and Q only depends on relative ratio λ therefore changes in constants of X_i do not affect the value of P and Q . Also, given that σ stays similar between experiments and thereby considered as constant, we can use the value of $P + Q$ as threshold of detection limit in SNR ratio value, which is multiplies of s for regression. Now the equation 1 can be rewritten in SNR ratio, as follows:

$$Y_D = P + Q$$

As log of 0 cannot be calculated, we excluded result of 0 and used 0.005 to 100, which corresponds to 1 and 5.301 in X_i . With the given standard condition, R is 4.301, $\bar{\lambda}$ is 0.554, N is 6 and the number of measurements is 10 at least. Therefore, with t for 0.98, i.e. $\alpha = \beta = 0.02$, at degree of freedom (4) = 2.999, the value of $P + Q$ is 4.446. The standard result is calculated with a regression curve;

$$SNR \text{ value} = k_{\text{sensor}} \times X_i$$

When $\alpha = \beta = 0.02$, the threshold SNR value is 4.446, the detection limit x_D for our ddFP biosensor is 0.011 ng ml^{-1} and $24.946 \text{ ng ml}^{-1}$ for FRET sensor (Supplementary Fig. 3)

All of the changes were made in the manuscript (page 5), supplementary figure 3, and methods.

4. Acronyms like RFU and HFU should be defined with full names.

As the reviewer suggested, we have described full names for the stated acronyms in the corresponding figure legends (Fig. 1, 2, Supplementary Fig. 1, 2, 4, 5, 7, 9, 10, 12).

5. In Fig.4, why are Ras signal changes highly variable? What are the possible reasons that high intensity puncta show a smaller signal change in the awake state? The tendency difference between awake state and anesthetized state seems negligible.

In awake behaving animals, many synapses are activated with different strengths. Some synapses are activated strongly and some synapses are activated weakly. Because of that, when we began imaging, it makes sense that some synapses have high Ras activity signals but some synapses have weak signals. During the continuous imaging sessions (30 minutes), it is presumed that some synapses receive continuous stronger inputs but some synapses do not. Conversely, some synapses that have been activated weakly now can get stronger inputs. Thus, the strength of synaptic inputs are kept changing.

Since we detected smaller changes at high intensity puncta, we postulated that those puncta represent synapses with continuous strong inputs. If synapses receive continuous strong synaptic inputs, it is likely that those synapses show stable Ras-signals with high intensity. If such synapses cannot receive continuous strong inputs due to the reduced neuronal activities in anesthetized animals, it is likely to observe large intensity changes (reduction) at synapses, which is consistent with our data (Fig. 4f).

Thus, we would like to emphasize that many synapses in layer 2/3 pyramidal neurons in the motor cortex are activated by dynamic synaptic inputs of different strengths. Such specificity of synaptic inputs may result in changing Ras-signals at specific synapses. Using new sensors developed in this study, we demonstrated that ~60 % of Ras expressing puncta remains the same (Fig. 4e). In addition, the different strength of synaptic inputs can result in Ras signal changes at certain synapses (remaining 40 %) highly variable at the synapse level.

6. In Supplementary Fig. 1, it would be better to show the actual iRFP fluorescence image instead of white dotted line representation

In Supplementary Fig. 1a, we co-expressed GA(cytosol or PM-targeted) and B copies without any labeling with iRFP. We draw white dotted line in panel a to show how many cells are in the image because it was hard to recognize cells with dim fluorescence. We apologize to the reviewer for miswritten figure legend and have revised it to avoid any confusion.

7. For the experiments in Supp Fig.7, it would better to include another set of control with G-Rac1 and RA-HRas-IRES-Lyn-iRFP.

As the reviewer suggested, we have carried out experiment in which we co-expressed G-Rac1 and RA-HRas-IRES-Lyn-iRFP682 in MDA-MB-231 cells and monitored RA and GA signals over time. Consistent with the previous result, we could observe that a substantial increase of GA signal (Rac1) near the protruded area but no noticeable elevation of RA signal (HRas), further demonstrating the specificity of our sensor without noticeable cross-binding of B-CRIB_{Pak1} to RA-HRas. The changes were made in the manuscript (page 7) and Supplementary Fig. 8.

8. Analysis of variance (ANOVA) should be used when comparing the values from multiple samples, including Fig. 1df, Fig. 2h, Supp Fig. 3c, and Supp Fig. 10bc. Detailed explanations of statistical analyses should be better described. In addition, the use of s.e.m. or s.d. for error bar should be consistent.

As the reviewer suggested, we have changed s.d. to s.e.m. in Fig. 1c, d, e, f, g, Fig. 2c and Supplementary Fig. 1b, 2c, 4b, c, 5, 6b, 7, 9c, d to consistently describe error ranges for each result.

In addition, we applied the ANOVA analysis to the results in Fig. 1d, f and Supplementary Fig. 4c (Supplementary Fig. 3c in the previous version of manuscript) and we confirmed that the results were still statistically significant. In cases of Fig. 2h and Supplementary Fig. 10b, c, however, we think that analyses by *student's two-tailed paired t-test* are more proper to determine significance of the results by reasons described as following.

Fig. 2h shows the relative changes of green fluorescence of G-Cdc42 (left) and G-HRas (right) in dendrites in different time periods (Transient vs. Sustained) following high-frequency uncaging stimulus (HFU). In order to examine the changes in fluorescence intensity of each small GTPase (i.e. G-Cdc42 and G-HRas) in dendrites upon HFU at spines, statistics were calculated *relative to baseline*. For Transient changes, the average of fluorescence intensities from two time points (0 min and 2 min) was compared to the baseline fluorescence intensity. For Sustained changes, the fluorescence intensity from the last time point (20 min) was compared to the baseline fluorescence intensity. In the bar graph, baseline was not plotted (Y axis is % baseline) and *student's two-tailed paired t-test* was used when the fluorescence intensity was statistically examined by comparing with baseline. Each experimental manipulation (i.e. G-Cdc42 and G-HRas) was independent and compared with its own baselines. Because ANOVA is used to test for differences among at least three groups (multiple sample comparison), two-group case in this study (data at different time points compared to baseline data) can be examined by a *paired t-test*. Data from 6 independent culture preparations were used for each experimental comparison. All statistics were calculated across spines (G-Cdc42: average of 1.2 spines per cell, n = 26 spines, 22 cells; G-HRas: average of 1.2 spines per cell, n = 22 spines, 19 cells). Error bars represent standard error of the mean (s.e.m.) and significance was set at $P = 0.05$ (*student's two-tailed paired t-test*).

Supplementary Fig. 10b, c show the relative changes of spine size (b) and relative changes in fluorescence intensity of small GTPases in spines (c) in different time periods (Transient vs. Sustained) following HFU. In order to examine the size (b) and the small GTPase activity (c) of HFU-stimulated, Mock-stimulated, and Spontaneous spines, statistics were calculated

relative to baseline. For Transient changes, the average of tdTomato fluorescence intensities (spine volume) from two time points (0 min and 2 min) was compared to the baseline tdTomato fluorescence intensity (b) and the average of green fluorescence intensities (small GTPase activity; G-Cdc42 or G-HRas) from two time points (0 min and 2 min) was compared to the baseline green fluorescence intensity (c). For Sustained changes, tdTomato fluorescence intensity from the last time point (20 min) was compared to the baseline tdTomato fluorescence intensity (b) and the green fluorescence intensity (G-Cdc42 or G-HRas) from the last time point (20 min) was compared to the baseline green fluorescence intensity (c). In the bar graphs, baselines were not plotted (Y axis is % baseline) and *student's two-tailed paired t-test* was used when the fluorescence intensity was statistically examined by comparing with baseline. Each experimental manipulation (i.e. HFU, Mock, and Spontaneous) was independent and compared with its own baselines. Because ANOVA is used to test for differences among at least three groups (multiple sample comparison), two-group case in this study (data at different time points compared to baseline data) can be examined by a *paired t-test*. Data from 6 independent culture preparations were used for each experimental comparison. Error bars represent standard error of the mean (s.e.m.) and significance was set at $P = 0.05$ (*student's two-tailed paired t-test*).

Reviewer #2:

We thank reviewer #2 for the positive assessment of our work and the constructive remarks. According to these comments, we have made following changes and improved the manuscript.

Remarks to the Author:

Summary

In this paper Kim and colleagues present a new brand set of genetically encoded probes to evaluate changes in the activity of small GTPases. The approach these authors choose combines the use of red-shifted based sensors with blue-light optogenetic modules. Such strategy would allow to measure changes in the activity of small GTPases in a temporal and spatially manner. As proof of concepts authors presented probes developed to evaluate the activity of Ras, Rac1 and Cdc42.

The paper is correctly written and results presented are sound. However, there are a number of issues that need to be addressed to deliver a more convincing message to readers .

Specific comments:

1. In abstract section authors refers to small GTPases as enzymes. This is not correct, please changes enzymes by proteins.

According to the reviewer's comment, we have changed the term in the manuscript (Page 2).

2. At the beginning of the results section, authors would quickly introduce what are the effector domains chosen for each GTPases, explicitly indicating why they choose these domains over others

As the reviewer suggested, we have described the reason for using the specific effector domains to design GTPase biosensors in our revised manuscript (page 4,5). Among various effectors known to interact with small GTPases, we chose RBD_{Raf1} and CRIB_{Pak1} domains to generate Ras and Rac1(and Cdc42) biosensors, respectively. Their bindings are well-characterized, highly specific to active small GTPase and have been typically employed for design of conventional FRET-based small GTPase biosensors (ref 4,7). Therefore, we reasoned that change of reporter domains from cyan-yellow FPs to ddFP modules with fixed pairs of GTPase-effector domain would provide fair comparisons between two types of sensors in terms of sensitivity (Fig. 1e, Supplementary Fig. 3), kinetics (Supplementary Fig. 5), fold change (Fig. 1f, Supplementary Fig. 5) and spatial distribution of signal (Supplementary Fig. 6).

3. The Supplementary Figure 3C indicate in first panel that upon expression of iSH2, there is a significant increase of R.HRas activity which contradicts authors statements in the manuscript as follow: "We also confirmed that some activators, such as iSH2 (activator of endogenous PI3K) and Vav2, could increase fluorescence signals of both RRac1 and R-Cdc42, but not R-HRas". Please correct or clarify.

As the reviewer pointed out, we have corrected the description in the manuscript (Page 5-6).

4. Authors state that "Quantification of local signal distribution revealed that the localized

activity pattern of R-HRas and G-Rac1 showed no significant difference in comparison with RaichuEV-HRas and RaichuEV-Rac1 respectively". If this is true, what would be the benefit of using G-Rac1 or R-HRas instead of Raichu probes? It is clear for this reviewer there are some benefits, but the message conveyed by authors get blurred with this statement

As we described above (Comment #2), we demonstrated outperformances of ddFP-based sensors over FRET-based sensors by comparing them in various properties such as sensitivity, kinetics and dynamic range. In another aspect, since the local activity of small GTPases is known to be critical in various cell functions, we tried to examine if our biosensors correctly represent spatial distribution of small GTPase activity by using FRET sensors as positive control. As a result, we found that use of the ddFP module instead of CFP-YFP pair did not distort the spatial profile of small GTPase activity during random migration of MDA-MB-231 cells (Supplementary Fig. 6). Therefore, through the whole process of characterization, we conclude that our sensors are able to sensitively and robustly represent small GTPase activity without any change or loss of spatial information of activity. To avoid any confusion regarding this point, we have clarified our intention in the revised manuscript (Page 6).

5. I have some concerns regarding Figure 2b as follow. It is not clear where in neurons light-activation of TrkB was induced. It may be very helpful to describe more in depth the experimental setup to perform these experiments. Additional, figure 2b should indicate the site for light stimulation in neurons. Is it the same if stimulation is done in axon, cell body or other neurites? Were there some morphological changes in neurons induced after Ras local activity? Such information is very relevant because it is widely accepted that a local Ras activation precedes axon specification (Fivaz et al 2000)

In Fig. 2b, we stimulated the neuron with global light illumination. For clarification, we marked the mode of activation in the figure and legend (Fig. 2b). In addition, we have elaborated how we activated cells by light in the Materials and Methods section.

Regarding subcellular activation, we have carried out experiments where we illuminated light on cell body or neurites and simultaneously monitored Ras activity and cell morphology. Under local and persistent delivery of light on the peripheral of cell body, we could observe substantial increase of Ras activity near the illumination site accompanied by membrane protrusion (Supplementary Figure. 11a). Interestingly, we found that proximal dendrites showed growth of filopodia-like structures where Ras activity was selectively detected at the base rather than within the filopodia (Supplementary Figure. 11b, c), indicating spatially confined regulation of Ras activity by TrkB during this process of morphological change.

Since it is hard to determine axons of fully mature neurons without any immune-staining step, we tried to locally activate OptoTrkB at distal parts of neurites of cultured hippocampal neurons at early stage of differentiation (DIV-2) in which neurites have not been specified as axons or dendrites yet. Under local and persistent illumination of light on a distal part of neurite, we found spatially restricted activation of Ras and efficient elongation of the neurite. In contrast, non-illuminated neurites did not show any extension or increased Ras activity; rather, they were retracted along with decreased activity of Ras (Supplementary Figure. 12a). When the stimulation site was changed to other neurites, we observed the same effects of shifted balance

of Ras activity and selective elongation of neurites. In addition, when we illuminated light on cell body, the previously extended neurite dramatically shrank. Thus, these results indicate that during the process of selective neurite outgrowth, subcellular regions undergo a competition to achieve neuronal polarization probably through utility of limited pools of key molecules such as Ras or actin (Fivaz et al., 2008, Winans et al., 2016).

As the previous study reported that local activation of Ras plays an important role in axon formation (Fivaz et al., 2008), we analyzed kinetic correlations between Ras activity and growth cone extension under local stimulation of OptoTrkB. As a result, we found that activation of Ras was detected before initial response of neurite extension (Supplementary Figure. 12b, c, d) and temporal cross-correlation analysis revealed that Ras activation preceded neurite extension by 0.534-min during the light stimulation (2-min to 30-min) (Supplementary Figure. 12e). Altogether, we proved high compatibility of our red-shifted sensors with blue light-based optogenetic modules that allows us to achieve multiplexed analysis and investigate spatiotemporal roles of protein activity under light-driven perturbation on a certain biological process.

6. Please include HFU acronym spelling in the results section. This will enhance readability and comprehension by readers

As the reviewer suggested, we have added full names for the acronyms in the manuscript and legends to enhance readability (Page 9, Fig. 2).

7. Regarding Figure 4a, I believe that results presented may support authors idea that these probes are useful for in vivo studies. However, it is not clear why authors choose to monitor Ras at the single synapse level instead of Rac or Cdc42 which are better proxys of the morphological changes dendritic spines undergoes upon neuronal activity. The paper would be very much strength if authors provide evidences for Rac and Cdc42 at synaptic sites

As the reviewer pointed out, the primary aim of our in vivo studies was to demonstrate capability of our sensors to monitor small GTPase activity in the brain of awake animals with a high resolution, which has been hardly achievable by previous methods. In addition, as accumulating evidence suggests that Ras GTPases besides Rho GTPases are critically involved in the synaptic transmission and various mental disorders (Harvey et al., 2008, Murakoshi et al., 2011, Stornetta and Zhu, 2010), it would be valuable to get information how Ras GTPases are dynamically regulated in physiologically relevant contexts. To this end, as shown in Fig. 4, we found that Ras activity was clearly detected by G-HRas sensor at micro-scale precision. Interestingly, it turned out that Ras activity was spatially distributed as discrete puncta that represented highly variable initial levels and dynamic changes of Ras activity at individual spines, possibly due to dynamic synaptic inputs with different strengths. In addition, we found an increase in the puncta population of ‘up’ category (increased Ras activity) and a decreased in the puncta population of ‘down’ category (decreased Ras activity) while mice were behaving on the treadmill for 30 minutes. In contrast, in anesthetized mice, opposite effect on Ras regulation was observed, reflecting that Ras activity is differentially regulated in a brain-state-dependent manner.

Regarding Rac and Cdc42, we have generated AAV encoding G-Rac1 or G-Cdc42 sensor and performed experiments where we tried to monitor activity of Rac1 or Cdc42 in the brain of freely behaving mice according to the reviewer's suggestion. Even though overall fluorescence of G-Rac1 and G-Cdc42 was dimmer than G-HRas sensor, we obtained consistent results with Fig. 4; puncta population of 'up' category increased and puncta population of 'down' category decreased, whereas anesthetized state showed reversed outcome (Supplementary Fig. 14). To enhance the robustness and efficiency of Rac1 and Cdc42 sensor, future efforts such as engineering of ddFP modules to increase their brightness or stability should be required.

Altogether, we would like to emphasize that our sensors will provide a valuable means to directly address dynamic nature of small GTPase activity with a high spatiotemporal resolution in various in vivo models, not only for the neuroscience study but also for other fields of research including cancer and developmental biology.

REVIEWERS' COMMENTS:

Reviewer #1 (Remarks to the Author):

The authors have done a great job of addressing all my comments from the previous review of this manuscript. The revised manuscript is now much refined. I recommend that the revised manuscript be suitable for publication in Nature Communications.

Reviewer #2 (Remarks to the Author):

I consider authors effectively addressed comments raised during initial revision of this manuscript. The revised version is substantially improved, and additional experiments are convincing.